# Sensitization to Horse Allergens—Molecular Analysis Based on the Results of Multiparameter Tests

**DOI:** 10.3390/ijms26041447

**Published:** 2025-02-09

**Authors:** Tomasz Rosada, Kinga Lis, Zbigniew Bartuzi, Natalia Ukleja-Sokołowska

**Affiliations:** Clinic of Allergology, Clinical Immunology and Internal Diseases, Ludwik Rydygier Collegium Medicum in Bydgoszcz, Nicolaus Copernicus University in Toruń, 87-100 Toruń, Poland; medtom@op.pl (T.R.); kinga.lis@cm.umk.pl (K.L.); zbartuzi@cm.umk.pl (Z.B.)

**Keywords:** allergy, horse, Equ c, molecular test, ImmunoCAP ISAC, Allergy Xplorer ALEX

## Abstract

The domestic horse is the third most common source of animal allergens. Currently, five equine allergens have been classified (Equ c 1, 2, 3, 4, 6). Despite the apparently low exposure to allergens, equine allergy is still of great clinical importance. The aim of the study was to analyze equine allergy based on the results of ImmunoCAP ISAC and Allergy Xplorer ALEX tests. The study was retrospective. A total of 1553 patients were analyzed. Immunological tests were performed using the ImmunoCAP ISAC and Allergy Xplorer ALEX systems. From all the results, those with a positive result for at least one horse allergen were selected for further analysis. Horse allergy was found in 9% of subjects by the ImmunoCAP ISAC and in 10% by the Allergy Xplorer ALEX system. In both tests, sIgE for Equ c 1 was most frequently found. Horse allergy was very often accompanied by allergy to other animals. Monosensitization to horse was found only in the ImmunoCAP ISAC tests (0.2% of the population). In the ImmunoCAP ISAC tests, a correlation was found between Equ c 1 and Can f 1, Can f 2, Can f 5, Fel d 1, Fel d 4, Mus m1 and Can f 3; with increasing sIgE concentrations for the allergen Equ c 1, the results for the remaining correlated allergens increased. Horse allergy is a common clinical problem. Equ c 1 is the major horse allergen. Monosensitization to horse is rare. The present study is the first to present analyses of sIgE concentrations for horse allergens.

## 1. Introduction

Animal allergens are the third most common cause of the development of allergic bronchial asthma, after house dust mites and pollen [1]. Cats and dogs, as the most popular pets, are also the most common source of allergization in this group. The animal species that comes next is the domestic horse (Latin: *Equus caballus*), a perissodactyl mammal from the equine family (Latin: *Equuidae*).

The great popularity of this species is associated with high exposure to its allergens, which are mostly classified as inhalant allergens (so-called aeroallergens). So far, four inhalant allergens of the domestic horse have been classified: Equ c 1–Equ c 4, and one protein classified as a contact allergen—Equ c 6 (lysozyme) [2]. Equ c 1 (lipocalin) is the major allergen in horses and is found in their saliva, coat and dander. The amount of allergen varies between breeds. The structure of Equ c 1 is similar to that of Mus m 1, suggesting that they may bind similar ligands. There is significant amino acid sequence homology between Equ c 1 and Fel d 4 (67%), as well as porcine lipocalin (61%) and rodent lipocalin (50%) [3,4,5]. Equ c 2 (lipocalin) is an allergen isolated from horse sweat. It has been found that sIgE to Equ c 2 is present in 33.3% of sera collected from people allergic to horses. It has about 50% similarity to the major bovine hair allergen Bos d 2 [5,6]. Equ c 3 (serum albumin) is the first isolated horse allergen to be confirmed in extracts of horse hair and dander. Mammalian serum albumins have a high sequence identity (72–82%), which increases the risk of cross-reaction between serum albumin allergens [5,6,7]. Equ c 4 (laterin) is a protein that plays a key role in thermoregulation and is one of the major allergens in horses; it reduces surface tension by facilitating the wetting of the horse’s coat, thus improving the flow of sweat for evaporation. In addition, it is likely that laterin also has bactericidal functions and significantly reduces the ability of microbes to adhere to horses’ hair [8,9]. Equ c 6 (lysozyme) is a protein with antibacterial properties that destroys bacterial cell walls. It is found in high levels in horse milk. It is classified as a contact allergen because it most commonly causes skin symptoms after the use of cosmetics containing horse milk. However, it should be remembered that the ingestion of this protein can also cause an allergy, in which case the symptoms are much more severe and can even lead to anaphylactic shock and death [10]. Potential cross-reactions between horse, dog and cat allergen molecules are shown in Figure 1.

Horse allergy primarily concerns people exposed to allergens of this animal; however, the literature describes cases of the development of clinical symptoms in the course of hypersensitivity to this animal but in the absence of obvious exposure to its allergens [12]. In these cases, it is postulated that there may be cross-reactions with allergens of other animals or so-called “hidden” exposure, i.e., contact with horse allergens, whose hair, partially altered in the production process, may be found in mattresses, furniture upholstery and the stiffeners (“lapels”) of jackets [13]. In the case of a horse allergy, symptoms mainly affect the respiratory tract due to the inhalant nature of most horse allergens. Allergic rhinitis and allergic bronchial asthma are most commonly observed, but allergic conjunctivitis is also diagnosed relatively often. Diagnosis of this type of allergy is difficult, as patients often do not associate exposure to this animal with the presence of clinical symptoms, which may be due to the fact that horses are overlooked as a possible source of allergy, the episodic nature of the exposure, the exposure to hidden allergens or, often, accompanying exposure to other, more probable in the opinion of patients and doctors, allergens during contact with a horse, such as house dust mites, grass pollen, tree pollen, etc. Most of the currently available allergy tests, i.e., skin prick tests (SPTs), specific IgE (sIgE) determination, allergen panels and molecular tests, can be used during the diagnosis of a horse allergy [11]. The treatment of choice is to avoid exposure to horse allergens. In the case of episodic contact that is not associated with the occurrence of severe clinical symptoms, pre-exposure prophylaxis can be taken into account, i.e., taking an antihistamine drug before the expected contact with the allergen and, after the exposure, washing thoroughly and changing clothes to limit further exposure. Individuals who, for various reasons, cannot limit contact with the animal may be subjected to specific immunotherapy, but this treatment should be preceded by thorough diagnostics, preferably molecular, and conducted under the care of an experienced allergologist [14].

The aim of this study was to assess the prevalence of sensitization to equine allergens in a multidirectional manner based on the results of molecular tests, for which patients were qualified on the basis of suspected allergy to various allergens.

## 2. Results

In the present study, sensitization to horse allergens was observed in 9% of the subjects in ImmunoCAP ISAC and in 10% in Allergy Xplorer ALEX tests, as shown in Figure 2.

The ImmunoCAP ISAC contains two horse allergen components: Equ c 1 (lipocalin) and Equ c 3 (serum albumin). In the presented results, sIgE for Equ c 1 was found in 83% of people allergic to horses, and that for Equ c 3 in 32%. In the Allergy Xplorer ALEX tests, horse epithelium extract Equ c_epithelia and Equ c 1 (lipocalin) were originally selected; however, in the second version of the test, modifications were made and, finally, three horse allergen components were included: Equ c 1 (lipocalin), Equ c 3 (serum albumin) and Equ c 4 (laterin).

Analyzing the obtained data, it was found that sIgE for Equ c_epithelia occurred in 63% of people allergic to horses, while the frequencies of sIgEs for individual horse allergen components were as follows: Equ c 1—75%; Equ c 3—25%; Equ c 4—16%. According to the accepted definition, the major allergen is the allergen that binds to IgE from serum in >50% of people who are sensitized to a given substance, so we would define only Equ c 1 as the major horse allergen. The above data are presented in Figure 3.

Polysensitization to other animals was very often observed among people allergic to horses. The percentage of sensitization to other animals was determined as follows: sensitization to cats—95%; sensitization to dogs—82%; sensitization to mice—40%; and sensitization to cows—31% in the group of 99 individuals allergic to horses by using ImmunoCAP ISAC tests (Figure 4). A similar profile of co-allergy was observed in case of using the Allergy Xplorer ALEX, where the percentage of sensitization to other animals was determined as follows: sensitization to cats—98%; sensitization to dogs—83%; sensitization to rabbits (40%); and sensitization to rats (30%) in the group of 40 individuals allergic to horses (Figure 5).

Analyzing individual allergen extracts and components, in the ImmunoCAP ISAC tests the most common were the sIgEs for Can f 1 (69%) and for Can f 6/Fel d 1/Fel d 4 (67% each), while those for Fel d 1 (80%) and Can f 6 (78%) dominated in the Allergy Xplorer ALEX tests. It is noteworthy that in the group of people allergic to horses, in ImmunoCAP ISAC study, sIgEs for allergenic components of other animals occurred in more than 25% of the subjects, while in the Allergy Xplorer ALEX tests, such frequency, i.e., min. 25%, was observed for the following components: Can f, Can f_Fd1, Can f_male urine, Can f 1, Can f 2, Can f 4, Can f 6, Fel d, Fel d 1, Fel d 2, Fel d 4, Fel d 7, Ory c 3 and Rat n. The above data are presented in Figure 6 and Figure 7.

Monosensitization to horses concerned a small group of subjects. In ImmunoCAP ISAC tests, only 0.2% of the examined population was allergic exclusively to horses, while in the Allergy Xplorer ALEX tests not a single such case was found. With regard to the allergen components, in the ImmunoCAP ISAC tests both the sIgEs for Equ c 1 and for Equ c 3 occurred with the same frequency—50%, as shown in Figure 8.

In the ImmunoCAP ISAC tests, some of the lowest average sIgE concentrations were recorded for horse allergen components—i.e., for Equ c 1, 7.33 ISU-E and for Equ c 3, 5.15 ISU-E (the lowest value among all the tested animal components)—and, thus, also the lowest sIgE levels among the subjects of the study. However, a correlation was found between Equ c 1 and Can f 1, Can f 2, Can f 5, Fel d 1, Fel d 4, Mus m1 and Can f 3, where, along with the increase in sIgE concentration for allergen Equ c 1, the results for the remaining correlated allergens rose as well, as shown in Figure 9. Also, in the Allergy Xplorer ALEX tests, the lowest values of mean sIgE concentrations were observed for Equ c_epithelia, 1.16 kU/L, and for Equ c 4, 0.5 kU/L, which were also the lowest sIgE levels among all the tested animal allergens. In this case, no statistically significant correlations were found between the evaluated sIgE concentrations.

## 3. Discussion

Allergy to equine allergens is the third most common allergy in the group of animal allergens. In the results presented, the percentage of allergic individuals was similar when using both the ImmunoCAP ISAC system and the Allergy Xplorer ALEX system. It should be noted that during data collection, the manufacturer of the Allergy Xplorer ALEX test changed the allergen composition of the horse molecules, i.e., Equ c_epithelia (allergen extract) was replaced by Equ c 3 and Equ c 4 (allergen molecules). The addition of the Equ c 3 molecule did not improve the diagnosis, as this allergen was never responsible for monosensitization and allergy to Equ c 3 always occurred together with allergy to another horse allergen. On the other hand, the addition of the Equ c 4 molecule seems to be a good solution, because among people who had sIgE to this molecule it was the only molecule of the horse to which sIgE was found in up to 60% of cases.

Horse epithelial extract (Equ c_epithelia) was only present in the older version of the Allergy Xplorer ALEX test. There was never a single-positive test result. In 37.5% of those allergic to horses, it was the only horse allergen for which sIgE was found. In a further 37.5% of those allergic to horses, sIgE was found for Equ c 1, while there was no sIgE to Equ c_epithelia. In 25% of the subjects allergic to horses, sIgE for the extract was present simultaneously with sIgE for Equ c 1. It appears that replacing the Equ c_epithelia allergen extract with two molecules (lipocalin and serum albumin) may improve the diagnostic value and does not affect the sensitivity of the test in determining hypersensitivity to horses.

The presence of monosensitization is so important in the presentation and analysis of the results because it represents cases of patients in whom an allergy to horses would not be found without the presence of sIgE to a specific allergen, indirectly confirming the validity of the presence of a specific allergen in the test.

The results of this study were in considered comparison with the results of previously published studies on horse allergy [13,15,16,17,18,19,20].

Arseneau et al. reviewed the literature available on horse allergy. They found that most analyses focus solely on cat and dog allergies, while horse allergy is often neglected. Despite the fact that horses are no longer used as the main traction force, the incidence of allergy to these animals is not decreasing at all, and this is probably connected with the wide use of horses for recreational, therapeutic and representative purposes. The authors reported that a surprisingly high percentage of allergies to horses also occur among city residents, where exposure to these allergens should be minimal. An interesting issue is the “hidden sensitization” in relation to horse allergy, the aim of which is to highlight the rare occurrence of clinical symptoms, despite allergy to horses, due to the rare exposure to the causative agent. The authors indicate only Equ c 1 (lipocalin) as the main allergen, which confirms the observation made in the presented study [13].

The frequent polysensitization among people allergic to horses is noteworthy. Allergy to dogs and/or cats was present in over 80% of people allergic to horses in both ImmunoCAP ISAC and Allergy Xplorer ALEX tests. This may result from the fact that the most common horse allergens belong to the groups of proteins with a highly conserved structure, i.e., lipocalins and serum albumins. The problem of cross-reactions in the group of animal serum albumins was examined by Spitzauer et al. They evaluated a group of two hundred patients, among whom 30% manifested hypersensitivity to albumins from various animals. Interestingly, patients diagnosed with hypersensitivity to horse serum albumin in skin prick tests (SPTs) also showed a positive reaction to at least one other animal albumin (dog, cat, pig, cattle, guinea pig, rabbit) [15]. Also, Cabañas et al., were interested in evaluating the importance of albumin as a cross-reactive allergen in patients allergic to cats, dogs and horses. They examined one hundred and seventeen patients allergic to cats, by means of skin prick tests (SPTs) and the RAST (radioallergosorbent test), for the occurrence of sIgE reactivity with extracts of cat, dog and horse fur and with purified albumin extracts of these animals. It was shown that 41% of patients allergic to cats were also allergic to dogs and horses; among them, 21% had sIgE reacting to the albumin from all three species (i.e., cats, dogs and horses), and 17% to albumins from two species. In the next stage, using an inhibition test, mutual inhibition was observed for albumins from cats, dogs and horses, but also for the hair extracts of these animals. The binding of sIgE to the horse extract was inhibited in 30% by the homologous albumin. Based on the obtained results, the authors concluded that cat, dog and horse albumins must have common epitopes which are responsible for the cross-reactivity occurring in about one-third of the examined patients. However, it was also shown that more than 50% of the sIgEs that cross-react in the case of cats, dogs and horses are directed against an allergen other than serum albumin (lipocalin?—author’s note) [16]. Cross-reactivity in the lipocalin group, on the other hand, was a subject of interest for Saarelainen et al. who conducted a study to confirm the occurrence of cross-reactions between lipocalins from different animals. Owing to the conducted inhibition tests, they documented the occurrence of cross-reactivity between Equ c 1 and Mus m 1 (house mouse allergen, lipocalin) and between Can f 1 and Can f 2, which allowed them to extrapolate that the relatively constant amino acid sequence and higher-order structures of lipocalins may be responsible for cross-reactivity between different animal species [17]. It is worth emphasizing that cross-reactivity does not explain all the correlations between allergies to horses and other animals, and polysensitization may also be of great importance. Further studies are certainly needed to clearly determine the relationships existing in this group of allergies. It is noteworthy that the results of studies by other authors cited above are in line with the results of our study.

The prevalence of sensitization to bovine and mouse components determined by the two different in vitro systems differed significantly. The differences observed in the case of cows may be due to the fact that each of the molecular tests discussed included a cow allergen molecule, but the manufacturers of the ImmunoCAP ISAC opted for Bos d 6, whereas the manufacturers of Allergy Xplorer ALEX opted for Bos d 2. As these are two different allergenic proteins, it is difficult to make comparisons. On the other hand, the differences found in the case of mouse allergy may be due to the different sensitivity and specificity of the two tests used in the study, as well as to the disproportions between the number of people allergic to equine allergens in each of the systems used.

In the present study, a very low percentage of monosensitization to horses was found. In the ImmunoCAP ISAC tests, only 0.2% of the study population were allergic to horses only, while in the Allergy Xplorer ALEX tests, not a single case was found. Similar results have been found in other studies as well. Liccardi et al. conducted a population study in Italy, during which skin prick tests (SPTs) were performed, among others, with an allergen extracted from horse hair. Among those with a positive result, a detailed medical history towards exposure to horse allergens was taken. In the group of people allergic to any of the allergen extracts tested, the percentage of allergy to horses was 5.38%. Significantly, only nine patients (i.e., 0.43% of those examined with any allergy) showed monosensitization to horse allergen extract. In the next stage of the study, it was found that among nine patients allergic exclusively to horses, six of them presented mild, periodic allergic rhinitis, and three others presented severe allergic rhinitis and allergic asthma. In addition, only three of the nine examined patients reported any contact with a horse (horse owners or riders), and the rest denied any contact with the animal [18].

Obando et al. evaluated the frequency of the occurrence of individual horse allergen components and the relationships between them. The study included 30 patients with a confirmed horse allergy. sIgE for horse allergens Equ c 1 and Equ c 3, the cat allergen Fel d 2 and the dog allergen Can f 3 were measured by means of ImmunoCAP^®^ and/or ISAC microarrays. Based on the obtained data, it was shown that 70% of patients had sIgE for Equ c 1 and 40% for Equ c 3. In 60% of the examined patients, monosensitization to Equ c 1 was found, and it was found in 30% to Equ c 3; only in 10% of patients were sIgEs for both tested horse allergens present. In addition, in patients allergic to Equ c 3 (horse albumin), 66% were allergic to Fel d 2 (cat albumin) and 58% to Can f 3 (dog albumin). Sensitization to Equ c 1 was connected with more-severe symptoms of allergic rhinitis (*p* < 0.001) and sensitization to Equ c 3 was connected with the persistence and severity of asthma (*p* 0.04, *p* 0.04, respectively) and prolonged rhinitis (*p* 0.01). Sensitization to both horse allergens (Equ c 1 and Equ c 3) was associated with the occurrence of a more severe form of allergic rhinitis (*p* < 0.01) [19].

The molecular profile of sensitization may be connected with a particular clinical picture of the allergy. Creating a specific set of “sensitivity profiles” and understanding the relationship between laboratory test results and the clinical form of sensitization is changing the way of thinking about allergic diseases and seems to be the future of allergology. It is worth noting that not only a specific “allergy pattern” but also sIgE concentrations and correlations between them may be of great importance in the final clinical picture of allergy. Gawlik et al. described a clinical case of a 6-year-old girl who suffered from anaphylactic shock after exposure to horse allergens. Interestingly, the sIgE concentration for the horse hair extract was >100 kU/L, and the girl was also allergic to cats and dogs, but it was not linked with the development of severe clinical symptoms that could directly threaten the child’s life, and in the performed tests, the sIgE concentrations for dog and cat hair extracts were many times lower (cat—2.43 kU/L; dog—5.95 kU/L) [20]. We can conjecture that the low sIgE concentrations for horse allergens that we observed may result from limited exposure and infrequent stimulation of the immune system in response to these molecules.

It is very important to observe the correlation between Equ c 1 and Can f 1, Can f 2, Can f 5, Fel d 1, Fel d 4, Mus m 1 and Can f 3, where the concentration of sIgE for the allergen Equ c 1 increased as the results for the other correlated allergens increased, especially since for most of the above-mentioned allergen molecules no potential for cross-reactions with Equ c 1 was found.

The diagnosis of equine allergy is not easy and requires great care and the right approach by the clinician. Accurate molecular diagnosis makes it more likely that a primary allergy can be distinguished from cross-sensitization and allows the selection of patients for whom specific immunotherapy may be particularly useful as part of causal treatment.

### Limitations of the Study

It should be noted that the tests on which the present study is based only allow the presence of sIgE to be detected, which makes it possible to determine sensitization in a given patient, but this is not the same as diagnosing allergy. In order to make a diagnosis of allergy, it is necessary to correlate the presence of sIgE with clinical symptoms following exposure to a particular allergen. The study was retrospective, so the data obtained are based on the analysis of results that were not originally intended for scientific evaluation and therefore lack many relevant clinical data. The results of the two tests, i.e., ImmunoCAP ISAC and Allergy Xplorer ALEX tests, cannot be directly compared as both tests were performed in a different study group, i.e., each participant had only one of the analyzed tests performed. This was mainly due to financial constraints, which did not allow both tests to be performed simultaneously on all participants. The study was conducted in a single center and involved patients from a specific region of Poland, which may also be important in terms of the frequency of exposure to individual animal allergens. The totality of these difficulties could certainly have influenced the results obtained, but efforts were made to make the above analysis as reliable and valuable as possible.

## 4. Materials and Methods

The study was retrospective in nature. The analysis included 1553 patients who, between 2012 and 2023, based on physical examination and medical history or other auxiliary tests, were suspected of having hypersensitivity to various allergens, both inhalant and food ones, and for this reason were qualified for molecular diagnostics based on multiparameter tests. Immunological tests were performed using two highly sensitive immunofluorescence methods: ImmunoCAP ISAC (Thermo Fisher Scientific, Diagnostics Austria GmbH, Wien) or Allergy Xplorer ALEX1/ALEX2 (MacroArray Diagnostics (MADx), Wien, Austria). At the time when the tests were performed, the manufacturers of both of them made modifications to the allergen composition, which was taken into account in the statistical analysis of the obtained results.

An sIgE concentration ≥ 0.3 ISU-E in ISAC’s ImmunoCAP tests and an sIgE concentration ≥ 0.3 kU/L in the Allergy Xplorer ALEX tests were considered elevated/positive, in accordance with common practice used in scientific research and in line with the recommendations of the test manufacturers.

Due to the subject of the study, out of all the test results received (the group joining the study), only those with a positive result, i.e., sIgE ≥ 0.3 kU/L or ≥0.3 ISU-E, for at least one horse allergen—i.e., in the ImmunoCAP ISAC: Equ c 1 and Equ c 3, and in the Allergy Xplorer ALEX tests: Equ c_epithelia, Equ c 1, Equ c 3 and Equ c 4—were selected and subjected to further statistical analysis. In the next stage of the study, the results obtained were also analyzed in terms of sensitization to allergens of other animals present in particular tests.

Therefore, 1553 patients with molecularly determined sensitization to various allergens were reviewed, and only those with sensitization to horse allergens were included in the study.

All the studies were carried out in the Immunology and Allergology Laboratory at the Clinic of Allergology, Clinical Immunology and Internal Diseases of Dr Jan Biziel, University Hospital No. 2 in Bydgoszcz. The methodology of the conducted immunological tests was in accordance with the manufacturer’s instructions, performed by an experienced laboratory diagnostician, in line with the standards appropriate for the particular procedure.

The consent to conduct the research was obtained from the Bioethics Committee of the Nicolaus Copernicus University in Toruń at the Ludwik Rydygier Medical College in Bydgoszcz (no. KB 297/2023 of 11/07/2023).

Graphical interpretation of the data was included in the form of vertical bar graphs and box and whisker plots. The correlation between two variables was calculated using Spearman’s R correlation coefficient. All calculations and figures were performed using Microsoft Excel spreadsheets and Statistica 10.0.

## 5. Conclusion

Horse allergy is relatively frequent, despite theoretically limited exposure to allergens. Equ c 1 is the main horse allergen; however, other allergen components also influence the frequency and the clinical form of the allergy. Monosensitization to horses is rare. Frequent polysensitization in the group of people allergic to horses is a very interesting observation, because it may be due to both simultaneous sensitization to several animals and cross-reactions between the allergen components. No analyses of sIgE concentrations in people allergic to horses have been published so far. The presented study is the first to show the above-mentioned relationships.

## Figures and Tables

**Figure 1 ijms-26-01447-f001:**
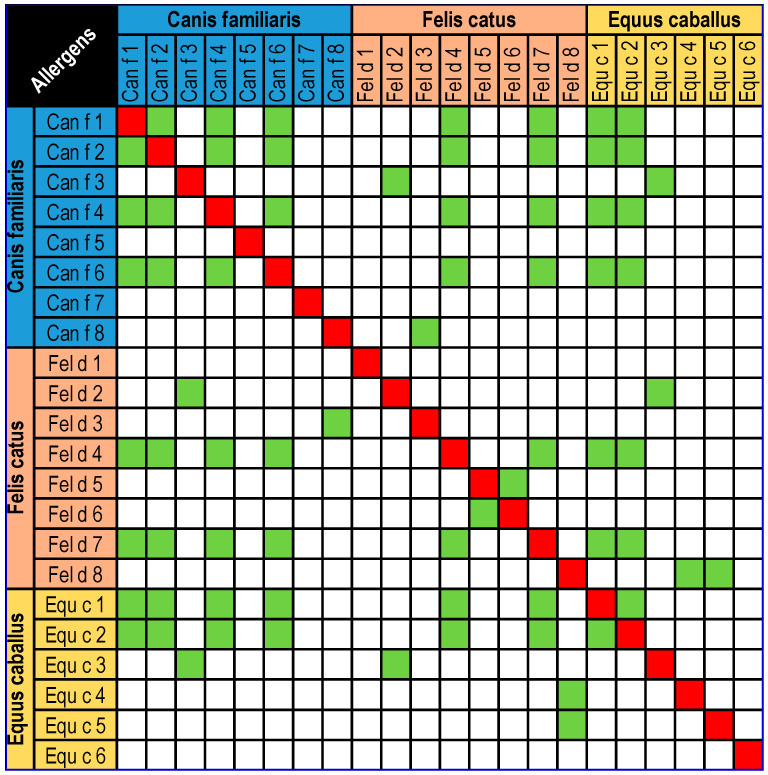
Potential cross-reactions between horse, dog and cat allergen molecules [11].

**Figure 2 ijms-26-01447-f002:**
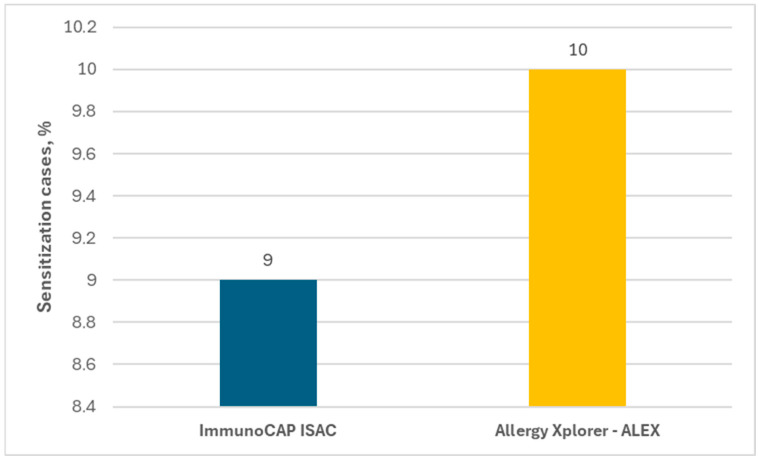
Percentage of people allergic to horses in each test.

**Figure 3 ijms-26-01447-f003:**
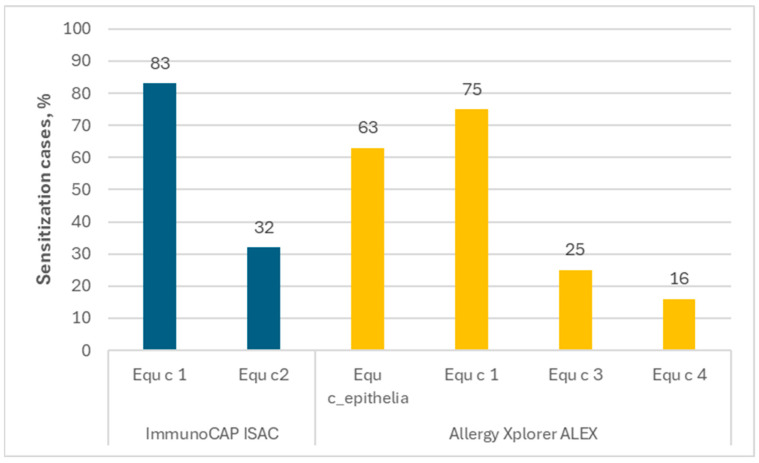
The frequency of the occurrence of sIgE for each horse allergen among people allergic to horses.

**Figure 4 ijms-26-01447-f004:**
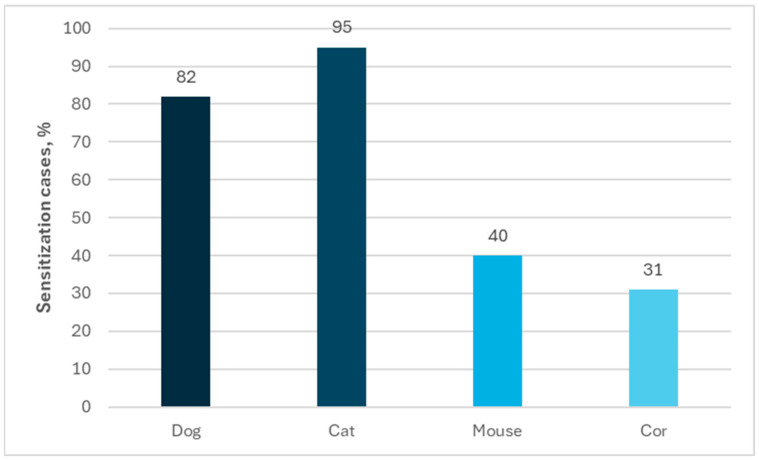
Percentage of allergies to other animals among people allergic to horses in ImmunoCap ISAC tests.

**Figure 5 ijms-26-01447-f005:**
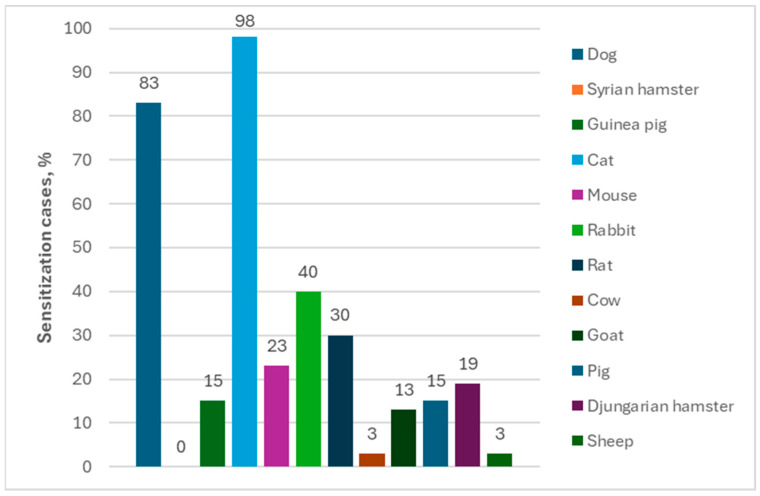
Percentage of allergies to other animals among people allergic to horses in Allergy Xplorer ALEX tests.

**Figure 6 ijms-26-01447-f006:**
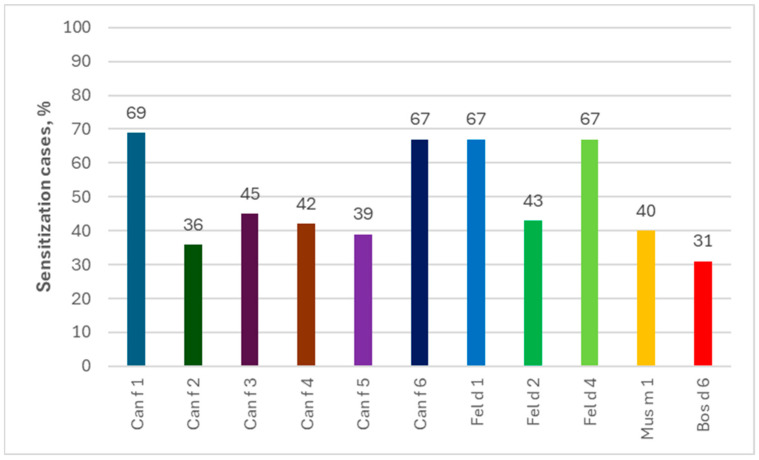
Percentage of allergies to components of other animals among people allergic to horses in ImmunoCAP ISAC tests.

**Figure 7 ijms-26-01447-f007:**
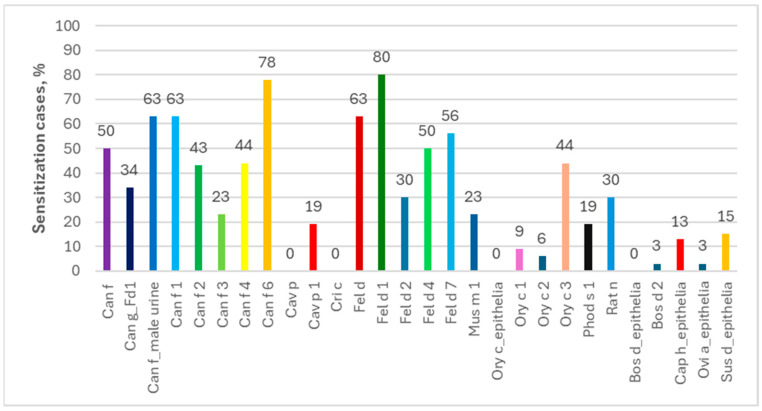
Percentage of allergies to components of other animals among people allergic to horses in Allergy Xplorer ALEX tests.

**Figure 8 ijms-26-01447-f008:**
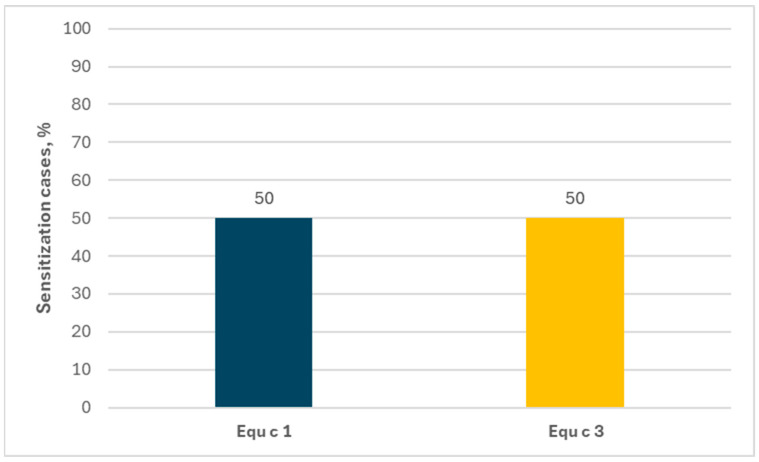
Percentage of allergen components among people allergic exclusively to horses (ImmunoCAP ISAC).

**Figure 9 ijms-26-01447-f009:**
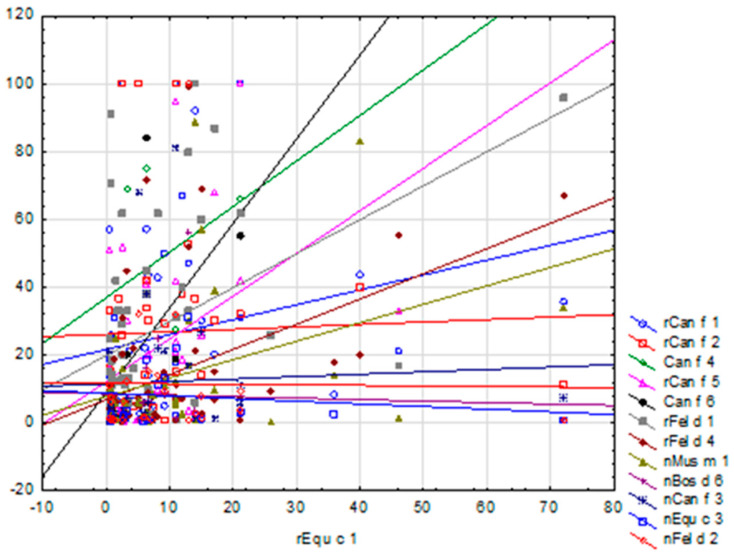
Scatter plot of sIgE concentrations for individual allergens against Equ c 1 allergen (ImmunoCAP ISAC).

## Data Availability

The data presented in this study are available on request from the corresponding author.

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
