# Peer review of "Sensitization to Horse Allergens—Molecular Analysis Based on the Results of Multiparameter Tests"

_ijms, 2025, doi:10.3390/ijms26041447_

Round 1

Reviewer 1 Report

Comments and Suggestions for Authors

The manuscript "Sensitization to horse allergens - molecular analysis, based on 2 the results of the ImmunoCAP ISAC and Allergy Xplorer – 3 ALEX multiparameter tests" by Rosada et al., is a retrospective analysis which included 1,553 patients (between 2012 and 2023) suspected of having hypersensitivity to various allergens (inhalant and food ones). The manuscript contributes to the molecular allergology of horse allergy. The manuscript is suitable to be published in the International Journal of Molecular Sciences and should be accepted for publication after minor revision.

Please in the introduction section give a brief overview of the structural features of horse allergens and eventual cross-reactivity with homologous allergens from other animal allergen sources. 

Please add producer, city and country for all chemicals/equipment ( such as ImmunoCAP ISAC and Allergy Xplorer - ALEX).

What was the reason for introduction asIgE abbreviation for specific IgE? It is usually sIgE.

Perhaps a table with cross-reactive horse, dog and cat allergens would be illustrative for the readers of the manuscript.

Author Response

Dear Reviewer 1,

Thank you for the time and effort it took to evaluate my Manuscript.

I am glad you found that the manuscript is suitable to be published in the International Journal of Molecular Sciences and should be accepted for publication after minor revision.

All your suggestions were used and now I feel that the Manuscript significantly improved.

I hope that in the current form you will find it suitable for publishing.

Kind regards

Natalia Ukleja-Sokołowska, on behalf of the authors

Reviewer 2 Report

Comments and Suggestions for Authors

General remarks

The manuscript is averagely written, although with a clear structure, clearly written material and methods and presentation of results.

General problem and the major concern is the structure of the discussion that includes only accidental, very short remarks on the results of the study (that are exaustively presented) that is highly insufficient. Majority of the discussion part is structured from the results and conclusions of other authors so it is a highly suspicios to be produced by AI and as a plagiat.

Cited References are completely missing, that is another sign we might be dealing with the AI product.

Author Response

Dear Reviewer 2,

Thank you for the time and effort it took to evaluate my Manuscript.

I improved the discussion as suggested, taking into account more detail description of the results in the light of current state of knowledge on horse allergy. I would like to strongly emphasize, that the manuscript was not written by any kind of AI, but our own original work, based on the results achieved in the Immunology Lab of our Department.

We feel, that, thanks to your insight, the Manuscript improved. I hope that after revision you will find that it is now ready for publication.

Kind regards

Natalia Ukleja-Sokołowska

Reviewer 3 Report

Comments and Suggestions for Authors I congratulate the authors on a great job, but I have a few observations. The manuscript, entitled “Sensitization to horse allergens - molecular analysis, based on the results of the ImmunoCAP ISAC and Allergy Xplorer – ALEX multiparameter tests“ describes the prevalence of sensitization to horse allergens according to one center experience. However, it is unclear from reading the paper why you presented the results of two different in vitro studies, as the paper does not analyze the differences in the sensitization profiles obtained by two in vitro techniques. Also, I have doubts about the calculations of correlations between the same family proteins in different animal species (for example between lipocalins: Equ c 1 and Can f 1, Can f 2, Can f 5, Fel d 1, Fel 19 d 4, Mus m1 and Can f 3) due to the crossreactivity. On the other hand, have you found that increased asIgE concentrations against horse allergens were associated with sensitization to other animals' allergen components? Comments: 1.      4 line:  “.“ is not needed in the title. Introduction 2.      The introduction is clear, concrete, and comprehensible, but does not state the manuscript's aim and objectives. Please highlight the aim of your work. Materials and methods 3.      I suggest clarifying the description of patients' inclusion in the study because the current description introduces uncertainty. Perhaps simply state that 1553 patients with molecularly determined sensitization to various allergens were reviewed and only those with sensitisation to horse allergens were included in the study. 4.      100 line: „The null hypothesis (H0) that there is no differ-100 ence in the study groups was accepted“ - what are these study groups? 5.      102 line: „so all P values below 0.05 were interpreted as indicating 102 the presence of significant correlations“ - redundant information. Results: 6.      Figures: please indicate the titles of the Y axes. I recommend ‘‘Sensitization cases, %“ and then not using ‘‘%‘‘for every value. 7.      Figure 1: Please check the Y-axis values and indicate the title of the Y-axis. 8.      Figure 2: please indicate the  title of Y axis. I also suggest clarifying the title of the figure, including <...> asIgE for each horse allergen component among people <...>. 9.      127-133 lines: I suggest rewriting the sentences: The percentage of allergies to other animals was determined as follows: allergy to cats – 95%, allergy to dogs – 82%, 128 allergy to mice – 40% and allergy to cows – 31%  in the group of 127 individuals allergic to horses by using ImmunoCAP ISAC"  instead of „In ImmunoCAP ISAC, the percentage of allergies to other animals in the group of 127 individuals allergic to horses was as follows: allergy to cats – 95%, allergy to dogs – 82%, 128 allergy to mice – 40% and allergy to cows – 31%“. The same with next sentence with ALEX, because the sensitization is determined by the tools, not "in" them. 10.  Figure 3,4,5, 6: The sensitization to cow or mouse components prevalence determined by the two different in vitro systems differed significantly. What could be the reasons for this? Please discuss this in the discussion section. By the way, I recommend not to use expressions such as „in Alex “ or „in ImmunoCap“ in the titles of the figures. Using the „determined by Alex/ImmunoCap technique“is better. 11.  Figure 8: it would be clearer if you made a table instead of a graph. Discussion: 12.  The first paragraph of „Discussion“ should be your work's main results and nowelty. There are only other researchers' results discussed in the discussion part. The discussion part format should be as follows: Your main results, the results obtained by other researchers, and then the explanation of the differences in the results. 13.  In short, the discussion lacks a deeper insight into the clinical relevance of the problem of sensitization to horse allergens. The current version of the manuscript only describes the prevalence of sensitisation. Two techniques were used to detect sensitization to horse allergens, where differences were obtained. It would be useful to explore these differences in more depth. Also try to explain why it is clinically important (or not) to determine the sensitization to horse allergens. The authors of the manuscript have done a great job, so it would be a disappointment that a weak examination of the results would diminish the value of your work. Conclusions 14.  Conclusions should be made from your results, so it would be beneficial to make it more narrow, not as abstract. 15. Expressions like: <...> is a very interesting observation [264-264 lines] are not recommended in scientific papers. References 16.  Only 16 references where 6 of them are self-citation.  The authors should include more scientific papers for comparison to get a broader picture of situation.

Author Response

Dear Reviewer 3,

Thank you for the time and effort it took to evaluate our work. Thank you for your kind words and positive comments. I read carefully all your suggestions and found them very helpful in improving our Manuscript. The revision was done point by point according to the review. I hope that in the current form you will find that the Manuscript improved and is now ready for publication.

Thank you for your help in making our work better.

Kind regards

Natalia Ukleja-Sokołowska

Reviewer 4 Report

Comments and Suggestions for Authors

I will suggest the authors to add some details and discussion on the clinical relevance of this study, as horse allergy is uncommon and people without contact with horse will not be expected to have clinical consequences.

For example, which are the people that need the test?

What if the allergy is detected among people that will not be exposed to the allergen?

Any issue of cross-sensitivity?

It will also be advisable to tabulate patient demographics.

Author Response

Dear Reviewer 4,

Thank you for the time and effort it took to evaluate our work. As suggested some details and discussion on the clinical relevance of this study was added. The problem of cross sensitization was described in detail. I feel that thanks to your insight the manuscript improved. I hope that in the current form you will find it suitable for publication. Once again thank you for your help in making our work better.

Kind regards

Natalia Ukleja-Sokołowska

Round 2

Reviewer 3 Report

Comments and Suggestions for Authors

Authors improved their manuscript but some corrections should be made:

86: I understood that your research aimed at determining the PREVALENCE of sensitization to horse allergens, not to compare results only after reading the limitation of your study. So it would be beneficial to make your aim more clear, for example The aim of the study was to assess the prevalence of sensitization to equine allergens in a multidirectional manner, based on the results of molecular tests, for which patients were qualified on the basis of suspected allergy to various allergens. Also, the Title of your manuscript introduces inaccuracy, because when you mention two specific systems, you are immediately assumed to provide a comparison of results. So my suggestion is not to mention the techniques used.  

Figures: it would be better than ‘’%’’ if you will write: Sensitization cases, % because the square brackets symbol [ ] means concentration. Also, it is not necessary to repeat the ‘’%’’ symbol after each value because it is indicated in the title of the Y axis.  

Fig. 1. Please check the Y axis (the values in each line are 9,9,9 and 10,10,10).  

149-155 lines: it is not correct to say ‘’allergy to animal”; it should be ‘’sensitization to animalbecause specific IgE is not necessary for allergy. Because Allergy is defined clinically, by symptoms on allergen exposure. A patient is considered sensitized when allergen-specific IgE (sIgE) antibody can be detected in serum or plasma or a skin test result is positive, even if no clinical reaction has been experienced. 

Figure 3,4. The title of the figure is not correct, according to my previous comment.  

22, 86, 206, 219, 281,329 lines: please changesensitiSationtosenzitiZation’. 

210-211 It is not accurate to say thatHorse epithelial extract (Equ c_epithelia) ,<....> was never responsible for monosensitization’’, because extract consists of various allergen components and we don't know if there are monosensitization (which means sensitization to one component) or more components.  

224 Please indicate references.  

342-345: Here again is the same problem: It should be noted that the tests on which the present study is based only allow the 342 presence of sIgE to be detected, which makes it possible to determine allergy in a given 343 patient, but this is not the same as diagnosing allergy. In order to make a diagnosis of 344 allergy, it is necessary to correlate the presence of sIgE with clinical symptoms following 345 exposure to a particular allergen. 

To determine senistization but not allergy. So, my suggestion is: ‘’It should be noted that the tests on which the present study is based only allow the 342 presence of sIgE to be detected, which makes it possible to determine sensitization in a given 343 patient, but this is not the same as diagnosing allergy. In order to make a diagnosis of 344 allergy, it is necessary to correlate the presence of sIgE with clinical symptoms following 345 exposure to a particular allergen’.  

References: It would be commendable if you included more recent literature

Author Response

Dear Reviewer,
thank you for your help in making our Manuscript better. I know it takes a lot of time and work and I appreciate it. I am sorry that we missed some of your comments during first round of review. We now prepared point by point review and all your suggestions were used.
I hope that in the current form you will find the manuscript suitable for publication.
Kind regards
Natalia Ukleja-Sokołowska